# Pathogen contingency loci and the evolution of host specificity: Simple sequence repeats mediate *Bartonella* adaptation to a wild rodent host

Ruth Rodríguez-Pastor[1☉], Nadav Knossow[2☉], Naama Shahar[2], Adam Z. Hasik[1¤], Daniel E. Deatherage[3], Ricardo Gutiérrez[4,5], Shimon Harrus[6], Luis Zaman[7], Richard E. Lenski[8], Jeffrey E. Barrick[3]*, Hadas Hawlena[2]*

1 Jacob Blaustein Center for Scientific Cooperation, The Jacob Blaustein Institutes for Desert Research, Ben-Gurion University of the Negev, Midreshet Ben-Gurion, Israel, 2 Mitrani Department of Desert Ecology, Swiss Institute for Dryland Environmental and Energy Research, Ben-Gurion University of the Negev, Midreshet Ben-Gurion, Israel, 3 Department of Molecular Biosciences, Center for Systems and Synthetic Biology, The University of Texas at Austin, Austin, Texas, United States of America, 4 National Reference Center for Bacteriology, Costa Rican Institute for Research and Teaching in Nutrition and Health (Incinesa), Cartago, Costa Rica, 5 Ross University School of Veterinary Medicine, Basseterre, St. Kitts and Nevis, West Indies, 6 Koret School of Veterinary Medicine, Faculty of Agricultural, Nutritional and Environmental Sciences, The Hebrew University of Jerusalem, Rehovot, Israel, 7 Department of Ecology and Evolutionary Biology, Center for the Study of Complex Systems, University of Michigan, Ann Arbor, Michigan, United States of America, 8 Department of Microbiology, Genetics, and Immunology, Michigan State University, East Lansing, Michigan, United States of America

☉ These authors contributed equally to this work.
¤ Current address: Institute of Evolutionary Biology, University of Edinburgh, Edinburgh, United Kingdom
* jbarrick@cm.utexas.edu (JEB); hadashaw@bgu.ac.il (HH)

**Data Availability Statement:** Raw sequencing data are available from the US National Center for Biotechnology Information (NCBI) Sequence Read

## Abstract

Parasites, including pathogens, can adapt to better exploit their hosts on many scales, ranging from within an infection of a single individual to series of infections spanning multiple host species. However, little is known about how the genomes of parasites in natural communities evolve when they face diverse hosts. We investigated how *Bartonella* bacteria that circulate in rodent communities in the dunes of the Negev Desert in Israel adapt to different species of rodent hosts. We propagated 15 *Bartonella* populations through infections of either a single host species (*Gerbillus andersoni* or *Gerbillus pyramidum*) or alternating between the two. After 20 rodent passages, strains with *de novo* mutations replaced the ancestor in most populations. Mutations in two mononucleotide simple sequence repeats (SSRs) that caused frameshifts in the same adhesin gene dominated the evolutionary dynamics. They appeared exclusively in populations that encountered *G. andersoni* and altered the dynamics of infections of this host. Similar SSRs in other genes are conserved and exhibit ON/OFF variation in *Bartonella* isolates from the Negev Desert dunes. Our results suggest that SSR-based contingency loci could be important not only for rapidly and reversibly generating antigenic variation to escape immune responses but that they may also mediate the evolution of host specificity.

Archive (SRA) under BioProjects PRJNA914220 and PRJNA1061316. Analysis scripts and source data are available at https://github.com/barricklab/ND-Bartonella-EE and https://github.com/barricklab/ND-Bartonella-SSR and have been archived in Zenodo at https://doi.org/10.5281/zenodo.10467299 and https://doi.org/10.5281/zenodo.10467301.

**Funding:** This work was funded by grants from the U.S.-Israel Binational Science Foundation (2017708 to H.H. and S.H.), the U.S. National Science Foundation (DEB-1813069 to L.Z., R.E.L. and J.E.B.), and the Israel Science Foundation (1391/15 to H.H.). R.E.L. was supported by a Hatch grant from the U.S. Department of Agriculture (7008075) to Michigan State University. L.Z., R.E.L., and J.E.B. received summer salary from the U.S. National Science Foundation grant. R.E.L. received academic salary from the U. S. Department of Agriculture grant. A.Z.H. received a stipend and support from the Zuckerman STEM Leadership Program. R.R.P. was supported by fellowships from the Kreitman School of Advanced Graduate Studies and the Swiss Institute for Dryland Environmental and Energy Research. The funders had no role in the study design, data collection and analysis, decision to publish, or preparation of the manuscript.

**Competing interests:** The authors have declared that no competing interests exist.

## Author summary

In nature, pathogens encounter a diverse range of host individuals and species. Understanding how pathogens respond to this high host diversity is essential for predicting and controlling infectious diseases in humans and wildlife. Does host diversity slow down and limit pathogen evolution or accelerate a pathogen's ability to access and adapt to novel hosts? Despite its importance, there are few experimental studies of how pathogens evolve in response to host diversity. To address this gap, we conducted a year-long laboratory evolution experiment to study how *Bartonella* bacteria from the Negev Desert dunes in Israel adapted to different native rodent species under low and high host diversity scenarios. Pathogen evolution did not proceed more slowly or quickly with the differences in host diversity. Instead, the pathogen rapidly adapted to the more challenging host species through mutations in mononucleotide repeats within an adhesion gene that is a virulence factor. Analysis of the genomes of *Bartonella* isolates from wild gerbils suggests that hypermutable repeats in this gene and others may have been selected and preserved by evolution, potentially enabling rapid and reversible adaptation to changing host environments. Our findings highlight a possible role for these "contingency loci" in the evolution of host specificity.

## Introduction

Parasites, including pathogens, face constantly changing host environments. Individual hosts change within their lifetimes, different host individuals exhibit variation, and some parasites infect multiple host species. How parasites adapt to these dynamic environments is difficult to observe directly. Models and experiments related to the evolution of specialist versus generalist parasites find that the availability of different host species influences the trajectory of parasite evolution, both in terms of the level of host specificity and the frequency of host shifts [1–5]. These dynamics, especially in vertebrate hosts, are often determined by an interplay between host immune factors and parasite antigenic elements [6]. For example, parasites may respond to changes in host availability by altering their surface features that are targeted by the host immune system, i.e., through antigenic variation [7]. However, only limited information is available about how parasites adapt to vertebrate host diversity in natural communities.

To begin addressing this gap, we studied *Bartonella* bacteria that are flea-borne pathogens of rodents living in sand dunes in the northwestern Negev Desert in Israel. This system offers the opportunity to investigate how parasites adapt in the context of a complex host community. In this region, a diverse collection of *Bartonella* strains circulates in communities with varying compositions of rodent host species including mainly *Gerbillus andersoni* and *Gerbillus pyramidum* [8,9]. *Bartonella* are prevalent in these rodent populations despite high strain turnover rates [10–12]. Infections with an individual strain are cleared by the host immune system, and that strain is subsequently unable to reinfect the same host [10]. This strong selection pressure from immune responses, coupled with the availability of multiple hosts, may favor *Bartonella* that have evolved the capacity to readily adapt to different host environments.

In line with this hypothesis, several genetic mechanisms that can facilitate rapid adaptation by generating antigenic variation have been noted in *Bartonella* species. First, *Bartonella* have high rates of intragenomic recombination events that copy, delete, and hybridize virulence genes with other nearby copies [13,14]. Second, *Bartonella* share a domesticated prophage that acts as a gene transfer agent, which can lead to recombination by exchanging DNA between

co-infecting strains [15]. Finally, it has been reported that mononucleotide simple sequence repeats (SSRs) of five or more bases are overrepresented in *Bartonella* genomes compared to other prokaryotes [16]. In other pathogens, genes containing SSRs have been shown to function as so-called "contingency loci" [17]. SSRs mutate at unusually high rates as a result of strand slippage during DNA replication, and these mutations often toggle expression of a gene between ON and OFF states [18]. When and how each of these mechanisms contributes to *Bartonella* adaptation is unclear.

To examine how *Bartonella* adapt to different host environments, we passaged a strain isolated from the Negev Desert dunes (originally sampled from a *G. andersoni* individual) through rodents of the two main host species. We passaged *Bartonella* through each host species individually and, in a separate treatment, alternating between each host species. We tracked *Bartonella* evolution by whole-genome sequencing and asked two main questions: (i) Do the mutation rates and targets depend on the history of encounters with the different host species? (ii) Do these mutations contribute to host adaptation? We found that mutations in SSRs in an adhesin gene dominated adaptation, arising in multiple lines independently and improving the ability of these *Bartonella* to exploit the more challenging (i.e., less favorable) host species. By sequencing the genomes of 38 *Bartonella* strains isolated in this region from wild gerbils, we found that similar SSRs are widespread and exhibit variation in their ON/OFF states. These results suggest that SSR contingency loci play an important role in allowing *Bartonella* to rapidly adapt to frequent host changes and thereby contribute to the remarkable local and global diversity of this genus of pathogens.

## Results

### *Bartonella* evolution experiment in rodent hosts

We evolved populations of *Bartonella krasnovii* OE-11 A2 for 363 days in three different rodent host scenarios: infecting *G. andersoni* only, infecting *G. pyramidum* only, or alternating between the two hosts (Fig 1). We started all lines from the same single colony isolate. For each of the three treatments, we started five independent lines and then passaged each line through 20 individual rodents by isolating and injecting bacteria into a naïve host. At each passage, bacteria were isolated from blood after 15 days of infection, and we used this population to inoculate the next rodent. Similar infection dynamics were observed across all passages and host treatments, and bacterial loads in infected rodents at 15 days post-inoculation were comparable across all passages (S1a Fig). Moreover, what little variation there was in the number of bacteria inoculated at different passages did not lead to major differences in final bacterial loads in the recipient rodents (S1b Fig). The absence of any change in bacterial loads in the alternating host environment across passages, compared to the single host environments, suggests that trade-offs across host species did not play an important role in *Bartonella* evolution during the experiment.

### Rates of genome evolution in different host treatments

We used whole-genome sequencing of 162 population samples and 147 endpoint clonal isolates to track new mutations that appeared during the evolution experiment (Fig 2). The overall genetic dynamics were consistent with selective sweeps by lineages that acquired new beneficial mutations, including several instances of lineages with different mutations competing and displacing one another (Fig 2a). Across all lines, we identified 27 unique mutations (S1 File), which included 17 base substitutions, 7 small insertions or deletions of 18 or fewer bases, 1 three-base substitution, 1 gene conversion that changed three bases in a 44-bp stretch to a homologous sequence found elsewhere in the chromosome, and 1 large deletion of 33,117

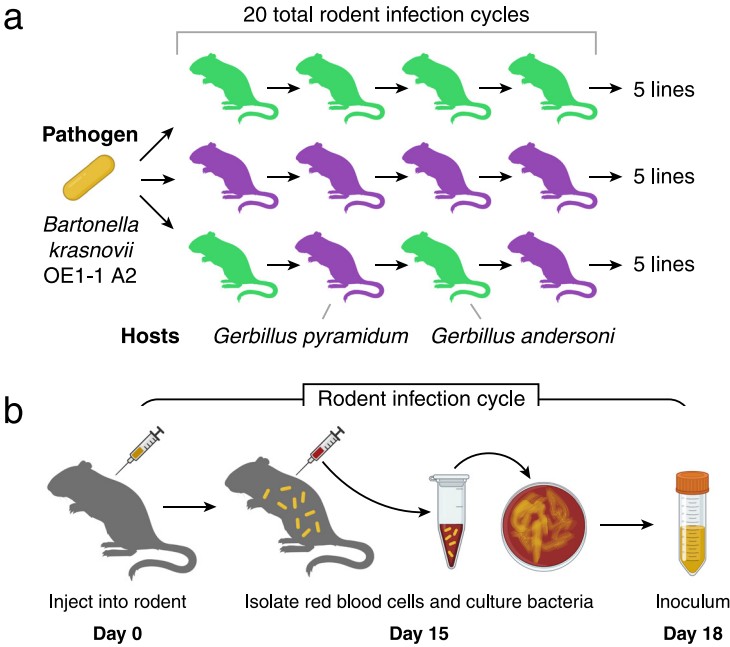

**Fig 1. *Bartonella* evolution experiment in rodent hosts. (a)** The design included 20 serial passages of *Bartonella krasnovii* OE1-1 A2 through individuals of one of the two host species (*G. andersoni* in green or *G. pyramidum* in purple) or alternating between them, with five independent rodent lines per treatment. (**b**) In each infection cycle, captive *Bartonella*-negative rodents were inoculated intradermally with bacteria. Bacteria were cultured from red blood cells sampled 15 days later to create the inocula for the next passage. A portion of each inoculum was archived for further study. Yellow rods and shading indicate bacteria in animal infections or samples. Figures incorporate artwork from Biorender.com.

bases. Of these mutations, 13 swept to high frequency and appeared to reach fixation in at least one experimental line, 2 were polymorphic in a final population (occurring in 2 or 8 of the 10 clones sequenced), 11 were found in only one sequenced endpoint clone and not observed in any population samples, and 1 was found only in population samples but not in a sequenced endpoint clone.

We found up to four new mutations in each of the final clonal isolates ([Fig 2b]). There were fewer mutations, on average, in an evolved *Bartonella* clone isolated from a population that was passaged only through *G. andersoni* (1.44 per clone) or only *G. pyramidum* (1.10 per clone) compared to a clone from a population passaged alternately through both rodent species (1.89 per clone). However, these differences among treatments in the rate of genome evolution were not statistically significant (repeated measures ANOVA: $F_{2,12} = 1.1$, $p = 0.37$).

## Recurrent and parallel SSR mutations occur in an adhesin gene

Indels in mononucleotide SSRs were the only successful mutations that recurred (independently arose in multiple lines) and exhibited genetic parallelism (with multiple distinct mutations affecting the same gene). Specifically, we detected changes in the lengths of two separate SSRs located within the first portion of the reading frame of the same trimeric autotransporter adhesin (TAA) gene ([Fig 2c]), which we named *badE* for *Bartonella* adhesin that evolved. TAAs that have been characterized in other *Bartonella* species are virulence factors that mediate binding or adhesion to host cells [19–21]. Within the *badE* open reading frame, we observed the deletion of a single A from a nine-base adenosine repeat in five populations, the deletion of

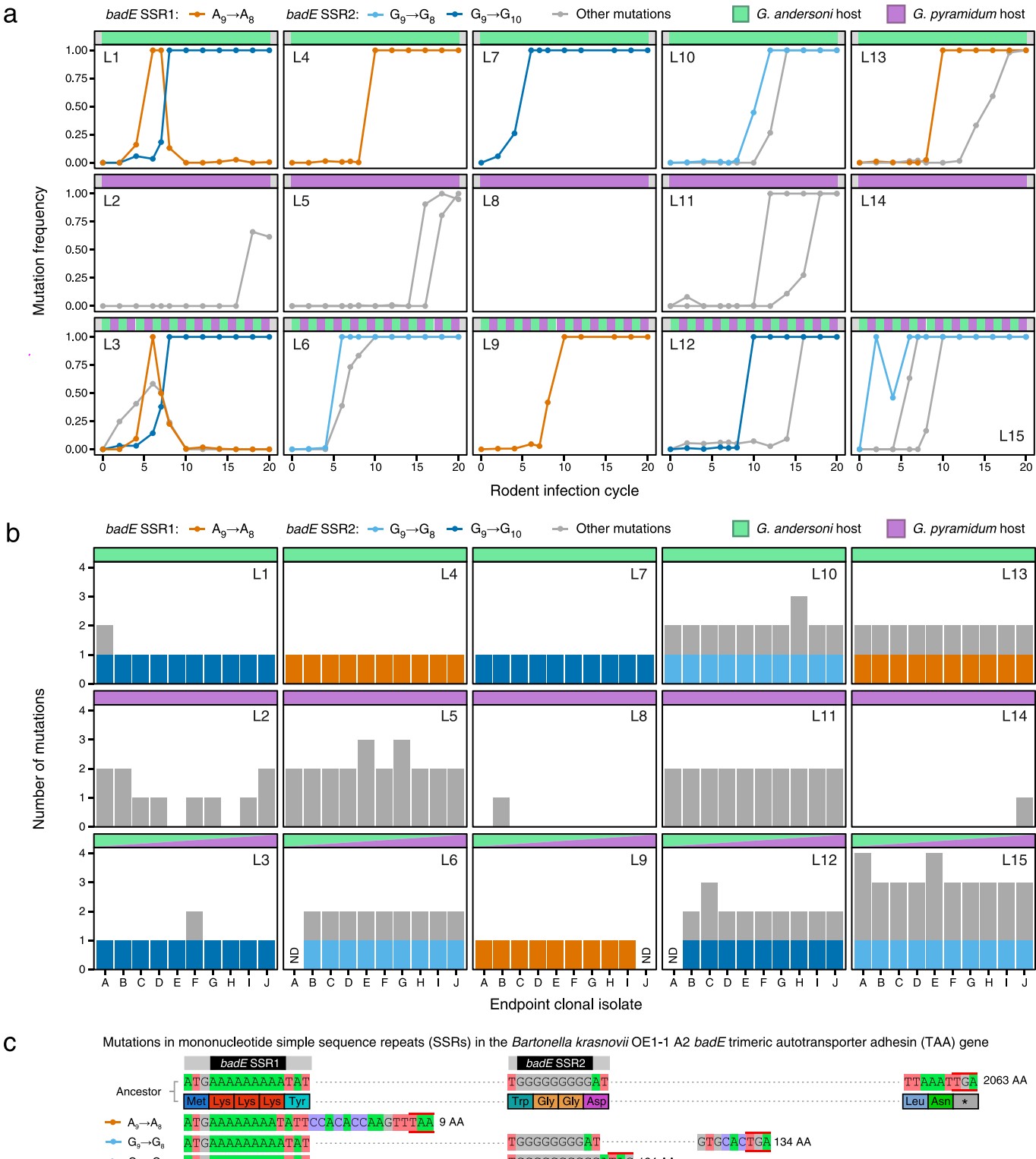

**Fig 2. *Bartonella* evolution is dominated by mutations in mononucleotide simple sequence repeats (SSRs) in the *badE* trimeric autotransporter adhesin (TAA) gene.** (**a**) Dynamics of competition between bacteria with new mutations in each of 15 experimental populations (L1-L15) over 20 rodent infection cycles. The relative abundance of each mutation was determined from metagenomic sequencing data. (**b**) Number of mutations in each of 10 clones isolated at the end of the evolution experiment from each of 15 populations. Each panel shows results for the five populations that were passaged only in *G. andersoni* (L1, L4, L7, L10, L13), the five passaged only in *G. pyramidum* (L2, L5, L8, L11, L14), or the five passaged alternately in the two rodent species (L3, L6, L9, L12, L15).

Sequencing data was insufficient to call mutations in the three clones marked ND (not determined). In **a** and **b**, mutations that altered the lengths of two mononucleotide SSRs in *badE* are shown in color. All other mutations are displayed in grey. For full details see S1 File. (**c**) Effects of *badE* SSR mutations. Key portions of the ancestral gene's nucleotide and amino acid sequences are shown in the first two rows. SSR mutations and how they lead to frameshifts and premature stop codons that inactivate the gene are shown in the three bottom rows. Stop codons are indicated by red bars above and below nucleotide sequences. Dotted lines represent portions of the reading frames that are not displayed.

a single G from a nine-base guanosine repeat in three populations, and the insertion of a single G in the same repeat in four populations. All three mutations result in frameshifts in *badE* that presumably inactivate its function.

These *badE* mutations occurred in 10 of the 15 lines, all of which experienced *G. andersoni* hosts. The SSR mutations increased in frequency until they were present in all or nearly all cells in these populations (Fig 2a), and they were found in all of the corresponding endpoint clones (Fig 2b). No mutations in these SSRs or elsewhere in *badE* were observed in any samples from the five lines that were passaged only in *G. pyramidum*. In some lines (L1 and L3), multiple mutations in the *badE* SSRs competed for dominance (Fig 2a). The genome of the ancestor of the evolution experiment has 85 other A/T homopolymer repeats and 13 other G/C homopolymer repeats of 9 or more bases, but no mutations in these other SSRs were observed during the evolution experiment.

We also re-analyzed sequencing data from a prior experiment that passaged populations of a closely related strain of *B. krasnovii* through 50 single-colony passages *in vitro* on agar plates [22]. There were no changes in the lengths of the same two SSRs in the *badE* gene under the conditions of relaxed selection in this experiment. Thus, hypermutability of these SSRs is unlikely to be sufficient for explaining why they consistently mutated in our *in vivo* evolution experiment. Taken together, the recurrence, temporal dynamics, and host-specificity of mutations in the two *badE* SSRs suggest that they improved the fitness of *Bartonella* in the *G. andersoni* host under the infection conditions of our experiment.

## Adhesin mutations affect infection dynamics

To test the effects of a *badE* SSR mutation under our experimental conditions, we compared the dynamics of infections of each rodent host species with the *B. krasnovii* OE1-1 A2 ancestor and an evolved clone in which the only observed mutation was a G insertion in the second *badE* SSR (Fig 3). The duration of infections was significantly longer for both strains in *G. pyramidum* than in *G. andersoni* (ANOVA results for infection duration: $F_{1,21} = 14$, $p < 0.005$), indicating that the latter host presents a more challenging environment for the pathogen. We found significant differences between the infection dynamics of the mutant and ancestor strains, with a faster increase in bacterial loads earlier in infections of *G. andersoni* but not *G. pyramidum* hosts (ANOVA, $F_{1,20} = 4.2$, $p = 0.05$ for host species × strain interaction and $p = 0.02$ for a planned comparison between the ancestor and evolved strains in *G. andersoni* at five days post-inoculation). This result confirms the expectation that this *badE* mutation is beneficial to bacterial fitness during infections of *G. andersoni*. Additionally, this mutation does not seem to negatively impact bacterial fitness during *G. pyramidum* infections.

## Evidence that *badE* is an SSR-mediated contingency locus

Our results suggest that the SSRs in the *badE* gene that mutated in the evolution experiment might have been selected and preserved by evolution so that this TAA gene can function as a contingency locus, in which high rates of SSR-mediated mutations can rapidly toggle gene expression OFF and ON within a population of cells through reversible frameshifts.

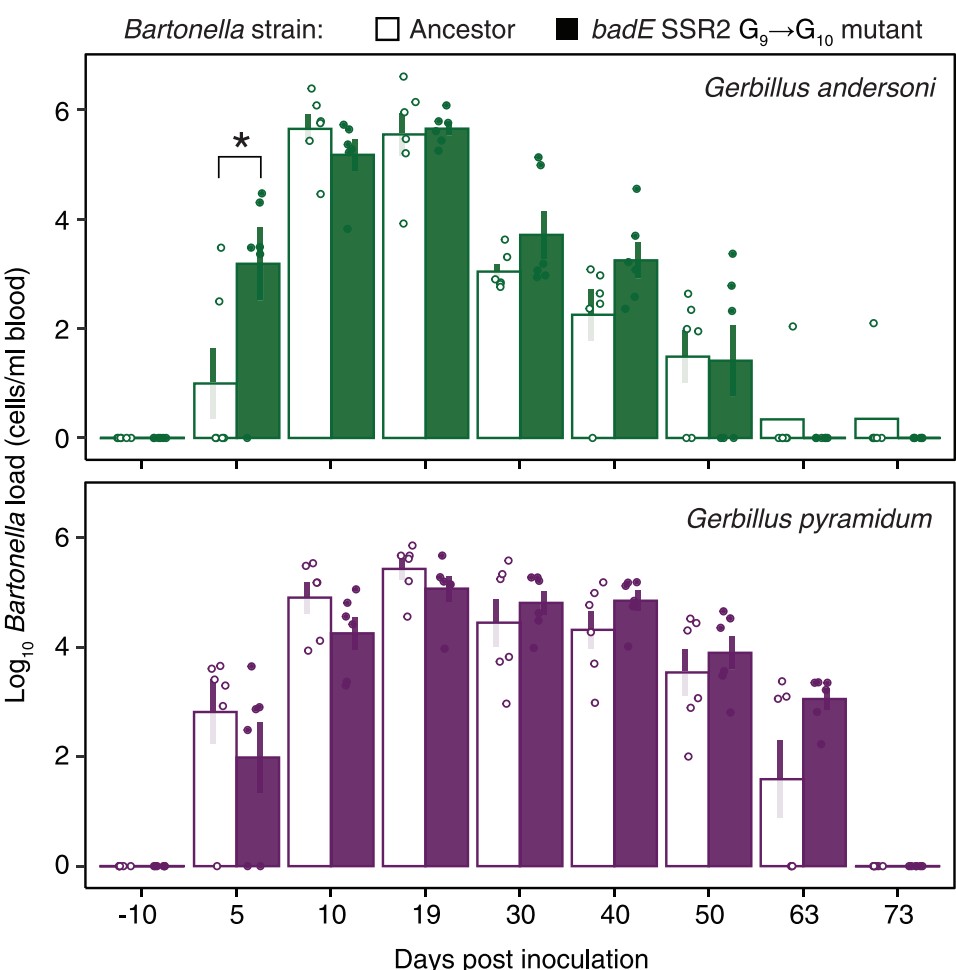

**Fig 3. Evolved SSR mutation in *badE* alters infection dynamics.** The *B. krasnovii* OE1-1 A2 ancestor strain or an evolved strain, differing only by a mutation in *badE* SSR2, were inoculated into three pairs of female and three pairs of male *G. andersoni* and *G. pyramidum* twins (one sibling was inoculated with the ancestor and the other with the evolved strain). Bars show mean ± s.e. of the log-transformed bacterial titer in blood samples collected on the specified days after inoculation. Points show values for individual hosts. *Significant difference between the mutant and ancestor groups. See the main text for statistics.

Alternatively, it is possible that the SSRs in the *badE* gene mediate evolvability in our experiment simply because they are mutational hotspots that happen to be there by chance. The hypothesis that the *badE* gene evolved to act as a continency locus would be further supported if: (1) the prevalence of hypermutable SSRs in its sequence was unusual and (2) the presence of these SSRs was conserved in related strains and species.

To examine the first question, we performed randomization tests in which we preserved the amino acid sequences of genes in the *B. krasnovii* OE1-1 A2 ancestor of the evolution experiment but shuffled the codons used to encode them within each reading frame (see Methods). Codon-shuffled versions of *badE* have 0.57 mononucleotide SSRs of nine or more bases, on average, compared to the two in the actual gene. This overrepresentation is only marginally significant ($p = 0.080$, one-tailed randomization test). However, this model does not take into account that there are typically fewer long mononucleotide SSRs than expected by chance in functionally important portions of microbial genomes [16,23–25], which has been attributed to selection against the genetic load of having hypermutable sites in essential genes [17]. In

line with this expectation, the protein-coding genes of *B. krasnovii* OE1-1 A2 have just 50 mononucleotide SSRs of nine bases or more compared to the average of 206 that are present in codon-shuffled gene sets, which is a significant depletion ($p < 0.001$, one-tailed randomization test). If we re-evaluate the SSRs in *badE* relative to this depleted baseline, then two SSRs is significantly more than one would expect ($p = 0.006$, one-tailed randomization test), and observing even one would be somewhat unusual (13.5% of codon-shuffled genes). In conclusion, there is some evidence that the *B. krasnovii* OE1-1 A2 *badE* gene has more SSRs than expected by chance.

To further examine the evolutionary conservation of *badE* and its SSRs, we sequenced and assembled the genomes of 38 *Bartonella* isolates collected in the Negev Desert dunes from wild gerbils. We identified and aligned the most closely related TAA gene sequences in each of these genomes and from eight representative *Bartonella* species that are not from this environment (Fig 4 and S2 File). Protein clustering based on sequence homology revealed that the *badE* TAA gene in *B. krasnovii* is more closely related to a different TAA gene in *B. henselae* than it is to the highly characterized *badA* TAA gene [19,21]. The *B. krasnovii badE* gene also does not exhibit close homology to the *Vomp* TAA genes of *B. quintana* [20].

The Negev Desert *Bartonella* are classified into four species groups that are closely related to one another and to *Bartonella* from other locations that also infect rodents. Of the 38 Negev Desert strains we sequenced, 18 are *B. krasnovii*, 14 are *B. gerbillinarum*, 3 are *B. khokhlovae*, and 3 are *B. negeviensis* [8]. The $A_9$ SSR that mutated in the evolution experiment encodes three consecutive lysines immediately after the start codon in the reading frame of the *B. krasnovii* ancestor's *badE* gene (SSR1 in Figs 2c and 4). This SSR is conserved in the TAA genes most similar to *badE* in all 18 *B. krasnovii* genomes. It is also present in the corresponding TAA genes in three of the four *Bartonella* species that are most closely related to *B. krasnovii*. By contrast, the $G_9$ SSR is present in only two strains: the OE1-1 A2 ancestor and OE1-1 G (SSR2 in Fig 2c). To put evolutionary conservation of the $A_9$ SSR into perspective, we examined an alignment of the 18 *B. krasnovii badE* genes and the most closely related *badE* gene from *B. elizabethae*. In this alignment, there are a total of 7762 nine-base windows, of which 3742 do not contain gaps. Only 358 of these (9.57%) have the same sequence in all 17 genes, as is the case for the $A_9$ SSR. In summary, we find that the $A_9$ SSR in *badE* is conserved within *B. krasnovii* and related species and that this is unusual but not exceedingly so compared to conservation of other sequences within this gene.

## Other candidate SSR contingency loci in Negev Desert *Bartonella*

To examine whether SSRs that might promote the evolvability of contingency loci were more widespread, we searched the 38 Negev Desert *Bartonella* genomes for protein-coding genes containing mononucleotide repeats of nine bases or more (S3 File). As was the case for *B. krasnovii* OE1-1 A2, each of these genomes contained roughly 50 such SSRs (Fig 5a). Adenosine repeats predominated over repeats of other bases on the coding strand (Fig 5b). They comprised 62–79% of all SSRs, depending on the species. SSRs were also noticeably concentrated at the 5′ ends of reading frames. We observed that 20–35% of SSRs were within the first 5% of the DNA sequence of the gene they overlapped (Fig 5c). This concentration of SSRs very early in genes was significantly greater than expected by chance in all four species ($p < 0.05$, one-tailed binomial tests). This trend also recapitulates what we observed in the *B. krasnovii* OE1-1 A2 *badE* gene where the $A_9$ SSR was immediately after the start codon and the $G_9$ SRR was within the first 6% of its length (S2 File).

Of the 181 different gene families that contained a mononucleotide SSR in at least two strains of the same Negev Desert *Bartonella* species, 41 were also found in at least two strains

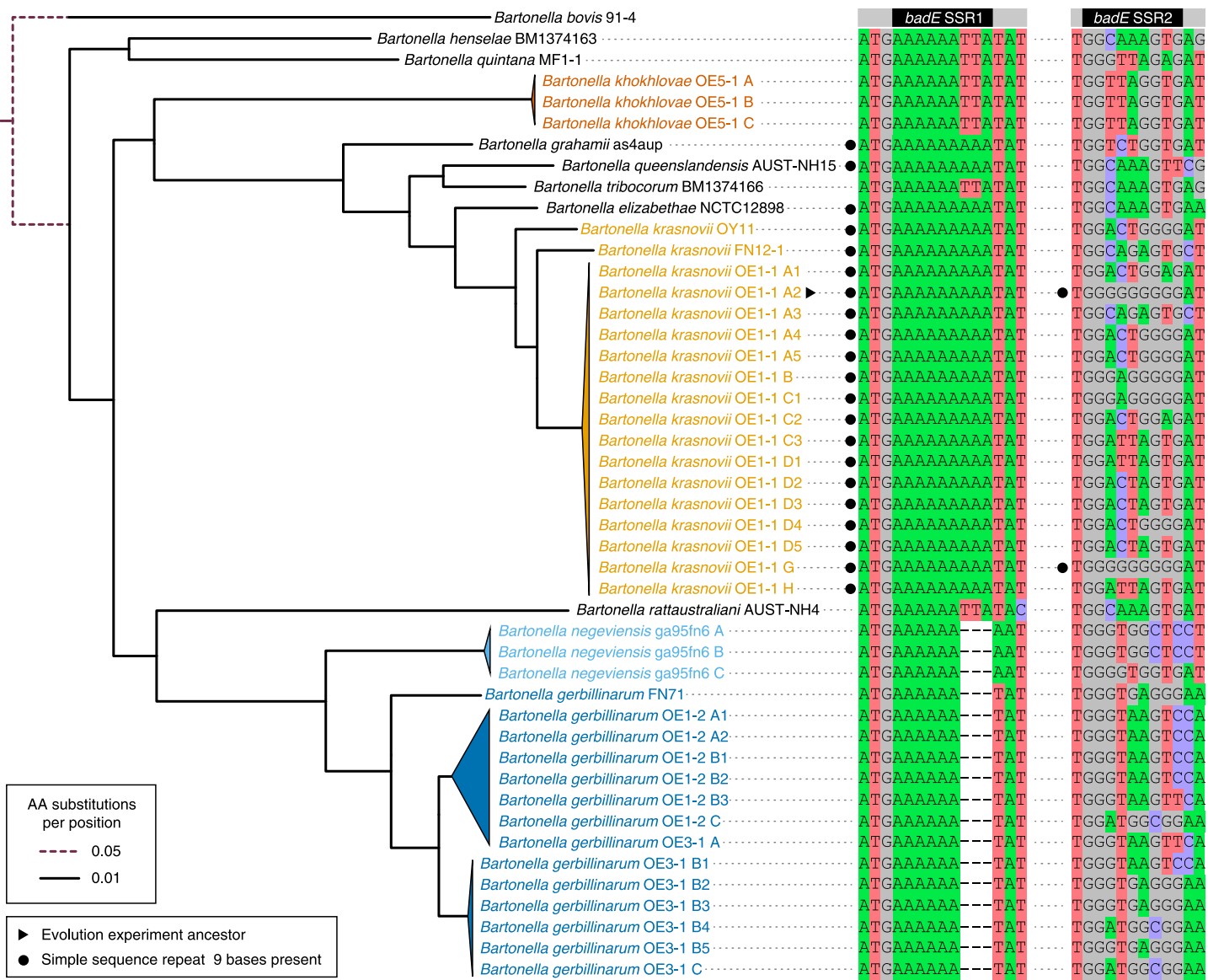

**Fig 4. Conservation of mononucleotide SSRs in TAA genes most similar to the mutated *badE* gene in Negev Desert *Bartonella* and their relatives.** The maximum likelihood phylogenetic tree of *Bartonella* strains on the left was constructed from a concatenated amino acid alignment of single-copy orthologs, with *B. bovis* 91–4 as the outgroup. Portions of the tree with very similar strains are collapsed into triangles. Every node in the tree not collapsed into a triangle was supported in all 100 bootstrap trees. The TAA gene in each *Bartonella* genome with the highest similarity to the first 200 amino acids of the *B. krasnovii* OE1-1 A2 *badE* gene that mutated during the evolution experiment was identified, and this region was aligned. The columns from this alignment that include or are adjacent to the two simple sequence repeats that mutated (SSR1 and SSR2) are shown on the right.

of one or more additional species (Fig 5d). Some of these conserved SSRs are in genes with functions related to the bacterial cell surface and virulence, which suggests they could operate as contingency loci for the evolution of antigenic variation or changes in host specificity (Table 1 and S3 File). For example, one adenosine repeat shared by all four Negev Desert *Bartonella* species is in a homolog of the lectin-like protein BA14k, which was first described as an immunoreactive protein in *Brucella* infections of rodents [26]. We did not see variation in the lengths of most conserved SSRs in the Negev Desert *Bartonella* that would lead to ON/OFF variation in gene expression due to frameshifts.

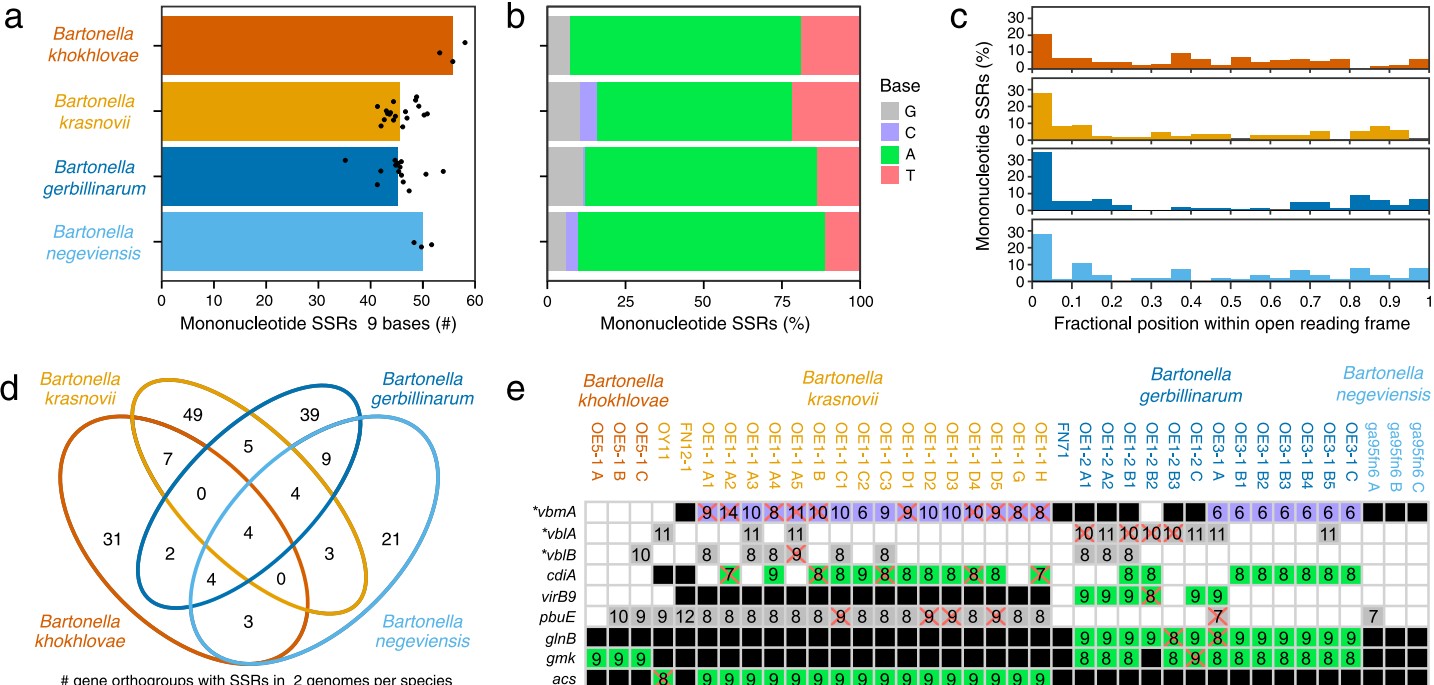

**Fig 5. Mononucleotide simple sequence repeats (SSRs) in the genomes of Negev Desert *Bartonella*.** (**a**) Number of mononucleotide SSRs with lengths ≥9 bases in the genomes of 38 strains of *Bartonella* belonging to four species that were isolated from rodents in the Negev Desert dunes (S3 File). Only SSRs overlapping protein-coding genes are included. Points are values for individual strains. Bars are averages within each species. (**b**) Fraction of mononucleotide SSRs in each species that are repeats of each base. Bases are specified on the coding strand of the gene containing the SSR. (**c**) Distribution of mononucleotide SSRs at different normalized positions within protein-coding open reading frames in each species. (**d**) Venn diagram showing the number of SSRs that occurred in at least two genomes from each of the four *Bartonella* species and how they are shared between these species. (**e**) Protein-coding genes exhibiting variation in the lengths of a mononucleotide SSR that result in frameshifts within the Negev Desert *Bartonella* strains. Numbers in boxes are the lengths of SSRs, with their backgrounds shaded by the repeated base as in **b**. A red X indicates a frameshift caused by the SSR in that gene. White boxes indicate that a gene was not identified in a genome. Black boxes indicate that the gene is present but does not have a mononucleotide SSR of ≥6 bases at that location. Asterisks indicate conserved genes of unknown function that were named in this study.

Other genes, however, contained SSRs that did exhibit length variation within strains from a single Negev Desert *Bartonella* species (Fig 5e, Table 1 and S3 File). These included a component of the VirB/VirD4 Type-IV secretion system (T4SS) [27], a T4SS effector, and three families of conserved *Bartonella* proteins of unknown function. We named one of these latter proteins variable *Bartonella* membrane protein A (*vbmA*). It contains a domain of unknown function (DUF6622), is predicted to have transmembrane helices, and has sequence similarity to NAD(P)/FAD-dependent oxidoreductases. The other *Bartonella* specific proteins with SSRs are from the same gene family. We named them variable *Bartonella* lipoproteins A and B (*vblA* and *vblB*) because they are predicted to be secreted by the Sec-system via signal peptides with cysteine at the cleavage site. Overall, these patterns of conservation and variation in SSRs (which are similar to the two SSRs that mutated during the evolution experiment) suggest that some could operate as contingency loci to rapidly and reversibly mutate to toggle gene expression between ON/OFF states.

## Discussion

To assess how differences in host species availability would influence parasite evolution, we passaged *Bartonella* populations through either a single rodent host species or alternating between two host species. We found that new genotypes with *de novo* mutations arose and

**Table 1. Highlighted genes with mononucleotide SSRs in Negev Desert *Bartonella* genomes.**

| Orthogroup§ | Gene# | SSR† | States‡ | Species¶ | Predicted function |
|---|---|---|---|---|---|
| OG0001164 | – | $A_{9-10}$ | ON | Bh, Bk, Bg, Bn | Lectin-like protein BA14k |
| OG0000503 | *lpxD* | $T_9$ | ON | Bk, Bg, Bn | UDP-3-*O*-acyl-glucosamine *N*-acyltransferase |
| OG0000315 | *clsC* | $T_9$ | ON | Bk, Bn | Cardiolipin synthase C |
| OG0000892 | *lolA* | $A_9, G_9$ | ON | Bk, Bn | Outer membrane lipoprotein carrier protein |
| OG0000423 | *pssA* | $A_9$ | ON | Bg, Bn | Phosphatidylserine synthase |
| OG0000008 | – | $C_{11}$ | ON | Bk, Bh | Autotransporter outer membrane β-barrel protein |
| OG0000618 | *yhhT* | $A_9$ | ON | Bg, Bn | Autoinducer-2 exporter (AI-2E) family protein |
| OG0001448 | *pbuE* | $G_{7-12}$ | ON/OFF | Bh, Bk | Purine efflux pump |
| OG0001520 | *shlB* | $T_9$ | ON | Bk, Bg | Hemolysin secretion/activation protein |
| OG0000099 | *cdiA* | $A_{7-9}$ | ON/OFF | Bk, Bg | T4SS effector, HNH/ENDO VII family nuclease |
| OG0000011 | *vblA** | $G_{10-11}$ | ON/OFF | Bk, Bg | Lipoprotein |
| OG0000011 | *vblB** | $G_{8-10}$ | ON/OFF | – | Lipoprotein |
| OG0001151 | *acs* | $A_{8-9}$ | ON/OFF | Bk | Acetyl-coenzyme A synthetase |
| OG0001185 | *glnB* | $A_{8-9}$ | ON/OFF | Bk | Nitrogen regulatory protein P-II |
| OG0000437 | *gmk* | $A_{8-9}$ | ON/OFF | Bh | Guanylate kinase |
| OG0001369 | *virB9* | $A_{8-9}$ | ON/OFF | Bk | T4SS system channel component |
| OG0000957 | *vbmA** | $C_{6-14}$ | ON/OFF | Bk | Membrane-localized NAD(P)/FAD-dependent oxidoreductase |

§OrthoFinder orthogroup into which *Bartonella* genes containing the specified SSR were categorized.

†SSR base and lengths observed in this gene.

‡Predicted state of gene function based on whether any examples of the SSR generate frameshifts.

¶Species in which the genomes of at least two strains contain the same SSR with a length of ≥9 bases. Abbreviations: Bh, *B. khokhlovae* Bk, *B. krasnovii*; Bg, *B. gerbillinarum*; Bn, *B. negeviensis*.

*Conserved *Bartonella* gene of unknown function named in this study.

outcompeted the ancestor strain in most bacterial populations by the end of the experiment. Mutations in two SSRs (simple sequence repeats) in the same *badE* TAA (trimeric autotransporter adhesin) gene dominated in lines that encountered one of the two rodent hosts. Evolution in changing host environments may favor different adaptive outcomes depending on mutation rates, bottleneck population sizes during infection, the strength of selection, the time spent in each host, the predictability of host shifts, and other factors. Our results provide insights into the evolutionary dynamics of *Bartonella*, specifically in the Negev Desert dunes, where these bacteria circulate within complex rodent communities.

Theoretical and empirical studies have shown that host diversity can either accelerate [28,29] or impede [30,31] the rate of parasite adaptation. Host diversity can also favor the evolution of parasites that are specialists or generalists, depending on the rates of parasite mutation and host switching and whether adaptation to one host results in trade-offs that reduce fitness in another host [32,33]. When the dynamics with which new parasite mutants arise and compete are slow compared to the rate of host switching, or when there are strong trade-offs, the evolution of generalists is favored, because any new parasite genotype must succeed in multiple host environments [34–36]. By contrast, when mutants that are more fit on one host can arise and dominate parasite populations before they switch hosts, and these mutants are less fit on other hosts, one expects parasites to evolve rapidly as they specialize on each successive host [32].

We found more mutations, on average, in the *Bartonella* lines that were passaged alternately through two rodent species than in lines that experienced only one host species. However,

these rate differences were not statistically significant because the rates were low and fairly uniform. More tellingly, we observed that multiple host passages were required for new mutations to increase in frequency and approach fixation within the evolving bacterial populations. This result indicates that the bacteria's evolutionary dynamics could not keep pace with the rate of host switching in the alternating host treatment. We did not detect strong trade-offs across the host species in the effects of successful mutations. Such trade-offs would have been expected to impede the spread of new beneficial mutations in the alternating host environment compared to the single host environments, which we did not observe.

These evolutionary dynamics suggest that our alternating host treatment selected for *Bartonella* that were better generalists, which is expected to proceed with adaptation first and foremost to the host that is initially less favorable [34–36]. It is likely that *G. andersoni* is a more challenging host for the *B. krasnovii* strain that we used as the ancestor. We found that *G. andersoni* infections have a shorter duration than *G. pyramidum* infections, and the latter species is the preferred host for fleas that transmit these *Bartonella* strains in nature [37]. This model also agrees with how SSR mutations in *badE* evolved in all ten lines that included *G. andersoni* hosts, either alone or as part of the alternating treatment, but in none of the five lines that faced only *G. pyramidum* hosts. The *badE* mutant strain also reached a higher titer in red blood cells than the ancestor early in *G. andersoni* infections. This difference in dynamics is relevant to our experiment because we isolated bacteria to continue propagating a line from each host 15 days after it was inoculated. Adaptation to *G. andersoni* apparently occurred without a trade-off, as we did not see changes in bacterial loads in infected *G. pyramidum* hosts during the evolution experiment, nor did we see a difference between the TAA mutant and ancestor strain during infections of *G. pyramidum*.

The experimentally evolved SSR mutations in *badE* cause frameshifts, leading to truncated products that are almost certainly nonfunctional. The *badE* TAA gene product in *B. krasnovii* is homologous to key virulence factors in other *Bartonella* species. These homologs include *badA* from *B. henselae*, which is necessary for infecting endothelial cells and inducing a proangiogenic response [19], and *Vomp*s from *B. quintana*, which mediate adhesion to collagen and autoaggregation [20]. However, the TAA gene family that acquired the SSR mutations in our evolution experiment is distinct from these characterized families. Though the function of this *badE* TAA gene family is unknown, it is likely to mediate binding to the mammalian extracellular matrix or otherwise affect interactions between bacterial and host cells [38]. TAA genes can be immunodominant antigens, and loss of Vomp expression has previously been observed during recurrent *B. quintana* bacteremia [20]. Loss of function mutations in the mutated *badE* TAA gene may be adaptive in some rodent hosts because they lead to different bacterial localization during infections or avoidance of the host's immune responses.

Future experiments are needed to determine the precise role of mutations in *badE* in enhancing *Bartonella* fitness within *G. andersoni* hosts. Nonetheless, our results suggest that SSRs acting as contingency loci could have a more prominent role in *Bartonella* adaptation and, more generally, in the evolution of host specificity than previously appreciated. We observed conservation of mononucleotide SSRs of nine or more bases in additional genes in the genomes of *Bartonella* from the Negev Desert dunes. These SSRs are concentrated at the 5′ ends of reading frames, as they are in other bacterial groups in which SSRs within genes function as contingency loci [25,39]. Furthermore, we detected length variation in some SSRs in *Bartonella* isolates from the Negev Desert, supporting the hypothesis that they evolved to mediate rapid ON/OFF switching of gene expression in nature. Some of the conserved and varying SSRs are associated with known virulence factors (e.g., type IV secretion systems and their effectors). Others are associated with conserved *Bartonella* genes of unknown function

that are predicted to be membrane-localized or secreted, which could mediate interactions with hosts and/or elicit immune responses. Additional experiments will be needed to understand if the TAA SSRs and SSRs in other genes act as contingency loci and their significance for pathogen evolution.

Our experiment reproduces some of the host complexity experienced by *Bartonella* in the Negev Desert, but it does not reflect other aspects of the natural environment, including transmission by fleas, co-infection of rodents with multiple *Bartonella* strains and species, and re-infection of rodents with immune systems that have cleared prior infections [8,12,40,41]. For example, mutations in the *badE* gene might be costly in the flea vectors or in some host species (e.g., *G. pyramidum*), which could explain why these mutations are less commonly seen in nature than in the evolution experiment. Fine-scale longitudinal sampling and genome sequencing of *Bartonella* in multiple wild rodent species and fleas could provide insights into the importance of SSR mutations under these conditions relative to other mechanisms that can promote evolvability, such as gene transfer agent-mediated recombination between co-infecting strains and species [15] and within-genome recombination in virulence gene arrays [13,14]. *Bartonella* bacteria possess many evolvability mechanisms that may contribute to the remarkable prevalence and diversity of these pathogens, underscoring the danger of zoonotic disease through spillover into humans.

The fixation of SSR mutations in our experimental *Bartonella* populations contrasts with the findings from an experiment that passaged *Campylobacter jejuni* in a mouse model [42]. In that study, no single SSR mutation reached 100% frequency in the evolved bacterial populations. This difference suggests there were more extreme population bottlenecks, less within-host replication, fewer beneficial pathways for SSR-based adaptation to follow, or some combination thereof during the *Bartonella* infections in our experiments. Overall, our findings broaden the typical view of the importance of SSRs in pathogen adaptation. Even when the rates of fixation of mutations at these loci cannot keep pace with host immune responses or movement to new hosts, they may provide an advantage for pathogens. Importantly, they can do so by generating focused genetic variation that allows for rapid adaptation to a challenging host species in a way that also leaves open the possibility of restoring the inactivated gene if it is needed for success in future host environments.

## Methods

### Ethics statement

Animal handling protocols were approved by the Committee for the Ethical Care and Use of Animals in Experiments of Ben-Gurion University of the Negev (permit number IL-76-09-2019B). Animal populations were held in the Hawlena laboratory with the permission of the Israel Nature and Parks Authority (permit number H3871/2019).

### Rodent hosts

All rodents used in the experiments were sourced from a laboratory colony managed by Hadas Hawlena. This colony is derived from wild rodents that have been born and raised in the laboratory under semi-natural conditions for approximately six years. These rodents have never been exposed to ectoparasites or any *Bartonella* species, nor have they undergone any drug treatments. The subjects of our study were all non-reproductive adults. They were housed individually in plastic cages ($34 \times 24 \times 13 = 10{,}608$ cm$^3$) on a 1-cm layer of autoclaved sand. The cages were located in an animal facility with an ambient temperature of $24.5 \pm 1^\circ$C and a 12-h light/dark cycle. The rodents had unlimited access to millet seeds and were provided alfalfa as a source of water.

## Evolution experiment

We serially passaged *B. krasnovii* strain OE-11 A2 (originally isolated from a *G. andersoni* host in the Negev Desert) either through 20 individuals of one of the two host species (*G. andersoni* and *G. pyramidum*) or by alternating between them (10 individuals of each host species). Each of the three treatments had five replicate lines. All lines were initiated from the same bacterial colony pick to remove initial genetic diversity. We also included a negative control line that was transmitted from one uninfected host to the next throughout the experiment and negative control hosts that were inoculated in each passage with phosphate-buffered saline (PBS). Neither control showed any evidence of *Bartonella* infection at any point. A full description of the protocol that explains the rationale for specific decisions (e.g., the ancestor strain, inoculation procedure, bacterial quantification methods, inoculation source, and day of sampling) has been published elsewhere [43]. Briefly, in each passage, *Bartonella*-negative rodents were intradermally inoculated with 100 μl of a PBS suspension of bacteria isolated from the previously infected rodent ($5 \times 10^6 \pm 5 \times 10^5$ total cells/inoculum). Rodents were bled by cardiac puncture 15 days after inoculation to collect *Bartonella* at the approximate peak of bacteremia. Red blood cells from each sample were spread on two chocolate agar (CA) plates. These plates were incubated for 3 days at 37°C and 5% $CO_2$. Then, bacterial cells were harvested by scraping the plates and resuspending the cells in PBS. This solution was homogenized by vortexing it with glass beads and passing it through a 5-μm filter. These samples were used to inoculate rodents in the next passage.

To assess bacterial loads in the rodent blood 15 days after each inoculation, we extracted DNA from the blood samples using a QIAamp BiOstic Bacteremia DNA Kit (Qiagen), following the manufacturer's instructions. To assess bacterial titers in the PBS suspensions used for inoculation, we extracted DNA using a thermal lysis procedure, as described previously [8]. In each set of extractions, we included a negative control, in which all the reagents were added to PBS but without adding the blood or bacterial suspension. We quantified bacterial loads by real-time quantitative PCR (qPCR), following Eidelman *et al.* [44]. The standard curve was created using DNA extracted from the ancestor strain and was calibrated to the count of colony-forming units of this sample.

## Genome sequencing

Bacterial populations isolated from rodents after passages 2, 4, 6, 7, 8, 10, 12, 14, 16, 18, and 20 were preserved from the PBS solution that remained after inoculating the next rodents by centrifugation, resuspending them in 1 ml of 20% (w/v) glycerol in lysogeny broth (LB), and then storing the samples at -80°C. DNA was extracted from these samples using the DNeasy Blood & Tissue Kit (Qiagen), following the protocol for gram-negative bacteria with slight modifications: centrifugation of the initial solution was at $5400 \times g$ for 10 min, and DNA elution was repeated twice in 50 μl of AE buffer that was preheated to 56°C before adding it to the purification column and incubating it for 5 min at room temperature. We also spread 100 μl of $10^{-5}$ and $10^{-6}$ dilutions of the bacterial solution that we cultivated from rodents after the last passage on CA plates. After 5 days of incubation at 37°C and 5% $CO_2$, we isolated 10 endpoint colonies for each experimental line. Each colony was propagated three times by streaking on a new CA plate and regrowing to ensure a clonal isolate. Following the last regrowth, 15 colonies were scraped from the plate for each clonal isolate and preserved in 500 μl of 20% (w/v) glycerol in LB. DNA extractions from these clone samples were performed as described above for the population samples.

In total, DNA was extracted from 318 samples: 165 populations (15 lines × 11 time points) and 153 clones (15 lines × 10 clonal isolates per population plus three replicates of the

ancestor). Up to 100 ng of purified gDNA from each sample was put into the 2S Turbo DNA Library Kit (Swift Biosciences). All reactions were carried out at 20% of the manufacturer's recommended volumes, with dual 6-bp indexes incorporated during the final 10-cycle PCR step. The resulting libraries were pooled and sequenced on an Illumina HiSeq X Ten instrument by Psomagen (Rockville, MD) to generate 151-base paired-end read data sets (S4 File). One population sample that was contaminated with DNA from a different *Bartonella* species was discarded before analysis.

## Mutation identification

We removed adaptors from the Illumina reads using *trimmomatic* (v0.39) in paired-end mode with the following settings: 4 allowed mismatches to the seed, a palindrome clip threshold of 30, a simple clip threshold of 10, and discarding trimmed reads ≤ 30 bases. To improve mutation calling, we assembled and annotated an updated version of the previously reported genome sequence of *Bartonella krasnovii* strain OE-11 A2 [45]. We used Trycycler (v0.5.3) [46] on nanopore long reads from a prior study (SRA:SRR6873507) [22] to generate a consensus assembly from Flye [47], Miniasm+Minipolish [48,49], and Raven [50] assemblies. Then, we polished the assembly by comparing it to our Illumina short reads for the ancestor strain using *breseq* (v0.36.1) [51] in consensus mode. Discrepancies between reads and the assembly were iteratively corrected by using the gdtools APPLY command and re-running *breseq* until no further improvements were found. Lastly, we used Prokka (v1.14.6) [52] to annotate the final assembly. The updated *B. krasnovii* OE-11 A2 reference sequence includes a chromosome with 2,163,391 base pairs and a plasmid with 29,057 base pairs (S5 File).

To call mutations in evolved clones and populations, we compared trimmed Illumina reads to the reference genome using *breseq* (v0.36.1) [51]. This pipeline identifies point mutations, small insertions and deletions, and structural mutations that can be predicted from split-read alignments, including insertion sequence (IS) element movements and large deletions. We first ran *breseq* in polymorphism mode with default parameters separately on each sample. Then, we combined the candidate mutations predicted by all *breseq* runs using gdtools MERGE and ran *breseq* again on each sample with the merged file provided as user evidence, so that it reported read counts supporting all candidate mutations in all samples. After this point, we did not further analyze samples that did not have ≥18-fold average read-depth coverage of the chromosome; one ancestor sample, two population samples, and three clone samples did not pass this cutoff.

Mismapping of reads to the reference genome, low coverage, and sequencing errors can lead to spurious mutation predictions. We employed several strategies to detect and disregard mutation predictions that are likely false-positives due to these or other biases. They rely on the following assumptions. First, mutations are expected to appear in clonal isolates with frequencies of either 0% or 100% and not at intermediate frequencies. Second, mutations should exhibit a frequency trajectory in population samples that begins at 0% and changes gradually over time in a correlated fashion, as subpopulations with the mutation outcompete others to reach high frequency or are outcompeted and diminish in frequency during the evolution experiment. Third, it is unlikely that the exact same mutation would be observed in all or most of the independent lines of the evolution experiment.

To implement these criteria, we began by keeping all mutations that were predicted at ≥90% frequency in any clone. Then, other predictions were examined for hallmarks of systematic errors of the types described above by analyzing all samples except for the ancestor controls. First, we disregarded mutation predictions in clonal or population samples that did not have at least 10 total reads supporting either the reference or mutated sequence

because estimates of their frequencies are subject to large errors. If >5% of all samples did not meet this threshold, that mutation candidate was not considered further. Then, we removed mutations that did not reach a frequency of 20% in at least one population sample. Next, we filtered out mutations that were observed exclusively in the population samples and were predicted to have a frequency of ≥5% in at least one population from 8 or more of the 15 different experimental lines. Additionally, we required the summed frequency of a mutation across all clone and population samples to be ≥100% for it to be considered further.

Finally, we fit a null Poisson regression model with a uniform rate of generating reads supporting the variant across all population samples. The offset in this model was the total number of reads that were informative about whether that mutation was present or not. For mutations supported by new sequence junctions the offset was corrected for the number of read start position registers that would lead to reads that could be definitively assigned to the junction. We compared this null model to a Poisson regression model that allowed for per-sample variation in the rate of generating reads with a mutation. If the likelihood ratio test comparing these two models was rejected after performing the Benjamini-Hochberg correction for multiple testing across all candidate mutations at a $10^{-6}$ false-discovery rate (indicating insufficient evidence for a mutation), then it was removed from consideration.

Mutation candidates that remained after these steps were manually examined and merged when they were linked by proximity and matching frequency time courses. For example, three nearby base substitutions that all seem to have resulted from one gene conversion event in line L11 were merged. The frequency of the merged mutation is reported as the average of the frequencies predicted for each individual mutation in these cases. Two mutations that passed all criteria peaked at frequencies of ~33% and ~25% in the population samples from lines L11 and L13, respectively, and these mutations were also found at roughly the same frequencies in the endpoint clonal isolates from these lines. These mutations may be in near-identical regions of the genome that were not individually resolved during assembly of the reference sequence. We corrected the frequencies of these mutations by dividing by the estimated peak value and capping the maximum frequency at 100%. For clarity, the frequencies of mutations were also assigned to be 0% in all samples outside of lines in which they were clearly present because they reached a frequency of >20% and did not exhibit sporadic jumps in frequency across population and clone samples. One mutation in L8 appeared at 100% frequency in all of this line's population samples and clones, indicating that it was already present in the progenitor cell that grew into the colony that was picked to initiate this population. Because this mutation did not arise during the evolution experiment, it was not counted or analyzed. Of the 27 mutations in the final list, 24 were at a >90% frequency in a clonal sample, 12 were predictions in population samples that passed the filters that eliminated systematic errors, and 9 were in both categories (S1 File).

## Infection experiment with *badE* mutant and ancestor strains

We inoculated three male twins and three female twins from each rodent species with either the ancestor or a mutant strain, and we compared their infection dynamics until the rodents cleared the infections. The mutant strain was a colony isolated after passage 20 from the L1 line, which was passaged exclusively in *G. andersoni* rodents. It was genetically identical to the ancestor except for a $G_9 \rightarrow G_{10}$ mutation in *badE* SSR2. We also had one control rodent of each species that was inoculated with PBS. To assess the rodents' bacterial loads, we bled them (50 μl of blood per sample) on days 5 and 10 post-inoculation and then every 10 days until day 73 post-inoculation. Blood was taken from the retro-orbital sinus under general anesthesia,

and a drop of local anesthesia (Localin, Fischer Pharmaceutical Labs, Tel Aviv, ISR) was placed in the eye. We used the same DNA extraction and qPCR protocols described in the section on the evolution experiment.

## Negev Desert *Bartonella* genome assembly and annotation

We assessed conservation and diversity in SSRs that may function as contingency loci across *B. krasnovii* OE-11 A2 and 37 additional *Bartonella* isolates classified into different genotypes. These isolates were cultured from fleas and blood sampled from 33 wild *G. andersoni* and 33 wild *G. pyramidum* rodents that were captured during October 2016 in two different sites in the northwestern Negev Desert sand dunes in Israel (34°23′E, 30°58′N and 34°23′E, 30°55′N) [8]. We performed DNA extractions and Illumina sequencing of these clonal isolates as described above for samples from the evolution experiment. Then, we assembled draft genome sequences using Unicycler (v0.4.9b) [53] on read subsets downsampled to 90× nominal coverage. Contigs that matched the genome of *Acinetobacter baylyi* ADP1 (GenBank: NC_005966.1) contaminated the initial 75C-4a isolate assembly due to barcode misassignment. They were identified and removed based on BLAST matches in Bandage (v0.8.1) [54]. Prokka (v1.14.6) [52] was used with the option to include Rfam predictions of noncoding RNAs to create the annotated assemblies that were analyzed. For all comparative analyses, we used the *B. krasnovii* OE1-1 A2 sequence assembled in this same way, rather than the closed assembly we used to analyze the evolution experiments.

## Evolution of *badE* SSRs in prior colony passage experiment

We re-analyzed data from a study that passaged a *B. krasnovii* A2-type strain that was isolated from a flea parasitizing a *G. andersoni* rodent in the Negev Desert through single-colony transfers on agar [22]. Illumina sequencing reads were downloaded from the NCBI Sequence Read Archive (SRP136159) and trimmed as described above. We assembled the genome of the ancestor of this experiment using Unicycler, as described for the Negev Desert *Bartonella*. Then, we identified the *badE* TAA homolog using a BLAST search and confirmed that it had both ancestral SSRs that mutated during our evolution experiment with *B. krasnovii* OE1-1 A2. Finally, we used *breseq* to call mutations in all nine of the clonal isolates that were sequenced at the end of the experiment after 50 colony passages. No mutations in either SSR were found in any of these samples.

## Codon randomization tests

Custom Python scripts relying on Biopython (v1.81) [55] were used to load protein-coding gene sequences from our closed assembly of the *B. krasnovii* OE1-1 A2 genome (S4 File) and perform randomization tests that kept the amino acid sequences of each protein in the genome fixed while randomly shuffling codons within each gene. This approach is similar to how prior studies have accounted for codon usage bias affecting SSR prevalence in microbial genomes [24,25]. We report 95% confidence intervals and *p*-values estimated from the characteristics of 10,000 shuffled versions of the *badE* TAA gene and 1,000 shuffled sets of all proteins in the genome. For comparing overrepresentation of SSRs in *badE* relative to the overall depletion of SSRs genome-wide, we performed the same procedure but added a step where each SSR in a shuffled gene had a 50/206 chance of surviving selection to make it into the final simulated distribution.

### Orthogroup identification and phylogenetic tree construction

To identify orthologous gene groups, we used OrthoFinder (v.2.5.5) [56] on proteins predicted in the genome assemblies of the 38 Negev Desert *Bartonella* isolates and representatives of 8 *Bartonella* species that infect other hosts. The phylogenetic tree was created by aligning the amino acid sequences of the 808 predicted single-gene orthologs with MAFFT (v7.508) [57], trimming these alignments with BMGE (v1.12) [58] using the BLOSUM62 substitution matrix, and inferring a maximum likelihood tree with IQ-Tree (v2.2.3) [59] from the concatenated alignment with 265,987 columns using a JTT+FO+R10 model and 100 bootstraps. We used *B. bovis* 91–4 as the outgroup to root the resulting tree.

### TAA gene family sequence analysis

We aligned the amino acid sequences of proteins assigned to the orthogroup containing both the TAA gene *PRJBM_RS00735* from *B. henselae* BM1374163 and the *B. krasnovii* OE-11A2 *badE* TAA gene using MUSCLE (v3.8.1) [60], with further manual refinement of the resulting alignment in AliView [61]. To recover incomplete reading frames interrupted by contig boundaries, we also added the best TBLASTN (v2.12.0+) [62] matches in each genome to the first 200 amino acids of the *B. krasnovii* OE-11A2 TAA sequence. To further delineate TAA families, we used InterProScan (v5.65–97.0) [63] to identify all proteins with matches to the head, stalk, and membrane anchor domains of YadA-like trimeric autotransporter adhesins (IPR008640, IPR008635, and IPR005594, respectively) in the 46 genomes that were analyzed. We aligned and clustered the sequences of these proteins and examined their placement relative to *B. henselae* BadA and *B. quintana* Vomp TAAs [19,20]. For the analysis of conserved amino acid triplets and nine-base subsequences, we used a custom Python script to examine a MUSCLE amino acid alignment of just the 18 *B. krasnovii badE* genes, which all have the $A_9$ SSR that mutated in the evolution experiment.

### Mononucleotide SSR survey

Mononucleotide repeats of ≥9 bases in the 46 *Bartonella* genomes were identified and associated with orthogroups using custom Python scripts that used Biopython and pandas [64] functions and R scripts that used tidyverse packages [65]. Orthogroups were assigned initial gene names and descriptions based on annotations in the input genomes, with further refinement through NCBI BLAST [62] and InterProScan [63] searches. We calculated overall statistics for SSRs that overlap protein-coding genes in the Negev Desert *Bartonella* species. Then, we further analyzed orthogroups with the SSRs found in at least two strains of any of the four Negev Desert *Bartonella* species (S3 File). We manually examined SSRs in sequence alignments for evidence of frameshifts caused by addition or deletion of the repeated bases using AliView [61]. For the *Bartonella* gene families of unknown function that contained SSRs with length variation, we predicted signal peptides using SignalP-6.0 (v0.0.56) [66] and transmembrane helices using DeepTMHMM (v1.0.24) [67].

## Supporting information

**S1 Fig. *Bartonella* infection loads and inoculum sizes throughout the evolution experiment.** We propagated *Bartonella* krasnovii OE1-1 A2 populations through rodents in three different host scenarios: infecting *Gerbillus andersoni* only (GA), infecting *G. pyramidum* only (GP), or alternating between the two hosts (GA-GP). (**a**) Mean of the log-transformed final *Bartonella* load in rodents from each host treatment at day 15 post infection, when blood was collected to culture bacteria for the next passage. (**b**) Log-transformed *Bartonella* loads in all

rodents from every passage plotted against the log-transformed size of the *Bartonella* inoculum injected into that rodent.
(EPS)

**S1 File. Evolution experiment mutations.** Mutations identified in 162 population samples and 147 endpoint clonal isolates from whole-genome sequencing data.
(XLSX)

**S2 File. *Bartonella badE* gene sequences.** Alignment of the *B. krasnovii* OE1-1 A2 *badE* TAA gene and the most similar genes in 45 related *Bartonella* genomes.
(FASTA)

**S3 File. Negev Desert *Bartonella* mononucleotide SSRs.** Characteristics of conserved SSRs found in 38 wild *Bartonella* isolates classified into four species.
(XLSX)

**S4 File. Genome sequencing data summary.** Characteristics of whole-genome sequencing data sets for the ancestor, 162 population samples, and 147 endpoint clonal isolates.
(XLSX)

**S5 File. *Bartonella* OE1-1 A2 reference genome.** Assembled and annotated reference genome used to predict mutations in evolved populations and clones.
(GBK)

## Acknowledgments

We acknowledge the Texas Advanced Computing Center (TACC) at The University of Texas at Austin for providing high-performance computing resources.

## Author Contributions

**Conceptualization:** Shimon Harrus, Luis Zaman, Richard E. Lenski, Jeffrey E. Barrick, Hadas Hawlena.

**Formal analysis:** Jeffrey E. Barrick, Hadas Hawlena.

**Funding acquisition:** Shimon Harrus, Luis Zaman, Richard E. Lenski, Jeffrey E. Barrick, Hadas Hawlena.

**Investigation:** Ruth Rodríguez-Pastor, Nadav Knossow, Naama Shahar, Adam Z. Hasik, Daniel E. Deatherage, Ricardo Gutiérrez, Jeffrey E. Barrick, Hadas Hawlena.

**Software:** Jeffrey E. Barrick.

**Visualization:** Adam Z. Hasik, Jeffrey E. Barrick, Hadas Hawlena.

**Writing – original draft:** Jeffrey E. Barrick, Hadas Hawlena.

**Writing – review & editing:** Ruth Rodríguez-Pastor, Nadav Knossow, Naama Shahar, Adam Z. Hasik, Daniel E. Deatherage, Ricardo Gutiérrez, Shimon Harrus, Luis Zaman, Richard E. Lenski, Jeffrey E. Barrick, Hadas Hawlena.

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
