## [Decision Letter · Decision Letter 0]

27 Aug 2024

Dear Prof. Barrick,

Thank you very much for submitting your manuscript "Parasite contingency loci and the evolution of host specificity: Simple sequence repeats mediate *Bartonella* adaptation to a wild rodent host" for consideration at PLOS Pathogens. As with all papers reviewed by the journal, your manuscript was reviewed by members of the editorial board and by several independent reviewers. The reviewers appreciated the attention to an important topic. Based on the reviews, we are likely to accept this manuscript for publication, providing that you modify the manuscript according to the review recommendations.

As you will see in the comments, reviewer 1 would like to see new experimental evidence while the other reviewers did not request this. We have recommended 'minor revisions' which would give you the oppotunity to address Reviewer 1 in the text of the manuscript. However, if additional experimental work is required to address these issues, then an extension can be granted to allow additional time to complete this work.

Sincerely,

Min Yue

Academic Editor

PLOS Pathogens

Francis Jiggins

Section Editor

PLOS Pathogens

Michael Malim

Editor-in-Chief

PLOS Pathogens

orcid.org/0000-0002-7699-2064

As you may see the comments from three reviewers, particularly reviewer 1, I would like to see new experimental evidences to address their concerns.

Reviewer Comments (if any, and for reference):

Reviewer's Responses to Questions

**Part I - Summary**

Reviewer #1: Ruth et al., investigated Bartonella adaptation in rodent through infecitons of Gerbillus andersoni or Gerbillus pyramidum, or alternating infection between the two species. Mutations in two mononucleotide simple sequence repeats (SSR) causing frame shifts (gene off) of Bartonella autotransporter TAA were identified after 20 passages in Gerbillus andersoni, but instead of in Gerbillus pyramidum. The authors also confirmed SSRs are conserved in other genes showing ON/OFF variation in Bartonella isolates that circulate in rodent communities in the Negev Desert, suggesting SSR-based contingency loci play roles in evolution of host specificity.

It is of great interesting to identified SSR mediated mutations of TAA which is host species specific tracking from almost 300 samples whole genome sequencing data. However, this manuscript falls short of defining a biologically-relevant role of the process they describe, as there is no convincing data linking their observations to a significant aspect of Bartonella adaptation.

Reviewer #2: In this manuscript the authors study within host adaptation of the pathogenic bacteria Bartonella, whose strains naturally infect different species of rodents in the Negev Desert.

Using experimental evolution within two different host species and whole genome sequencing of the bacteria, the authors aimed at testing if the mutation rates and the genomic basis of adaptive evolution depended on the host and on the history of host infections.

By performing sequential infections of a strain into rodents in the laboratory during 20 infection cycles, they find that mutations in single sequence repeats (SSRs) of an autotransporter adhesion locus (which they call gene badE) arise independently. This bonafide adaptive locus is common across passages in a specific rodent species and also in infections where the hosts alternate. The SSR mutations are not seen during in vitro passages in a mutation accumulation experiment performed previously, strongly suggesting that the cause of their emergence in the infection is selection within a specific host and not hypermutation.

Their evolution data suggests that SSRs in badE contributes to increased fitness of the strain in specific hosts, and this hypothesis was confirmed by further in vivo infections with the ancestor strain and with an evolved mutant clone.

Interestingly, the adaptive region identified in the laboratory experiments is polymorphic in nature, which suggests that the identified locus contributes to the adaptation of this bacteria to host shifts. Finally the authors put the results in a broader phylogenomic context.

The paper is clearly written, addresses an important question using well conducted experiments and several analysis methods. It provides great results to the field of host-pathogen interactions.

Reviewer #3: Rodríguez‐Pastor et al. presented a study of the molecular evolution in parasites during the adaptation to alternative hosts. They selected and carefully justified a host-parasite system involving bacterial parasite Bartonella krasnovii and two rodent hosts Gerbillus andersoni and G. pyramidum, sampled from wild communities in the Negev Desert in Israel. This system is a close simulation of real-world host-parasite coevolution. In their main experimental evolution analysis, they involved a single colony isolate of parasite, and three host scenarios (either or both hosts), and 15 days by 20 infection cycles on each of 5 lines. This experimental setting has sufficient power to address the question of interest. They identified a total of 20 unique mutations across the experiment, and noted the recurrence of frameshift mutations in the mononucleotide simple sequence repeats (SSRs) of the what they named as badE adhesin genes. They investigated the biological roles and evolutionary patterns of these mutations, and proposed that these mutations confer evolutionary advantages during the infection of G. andersoni -- the harder host of the two. They found genetic evidence from wild Bartonella isolates of the same region that support this hypothesis.

Overall, this is a high-quality study of genetic and evolutionary mechanism underlying host specificity of parasites. It is thoughtful, comprehensive and logically sound, addressing an important biological question with considerable success. The experimental design and the model system will inspire future researchers. I did not identify any notable weakness in the methods and results. Therefore, I endorse the publication of this manuscript in PLOS Pathogens.

There is only one limitation that I wanted to note to the authors. The study does not involve flea as an essential vector during the transmission of Bartonella parasites among rodent hosts. The authors briefly admitted this in the discussion. I think it is reasonable to guess that the mutations of the badE gene may have detrimental effects during the flea stage, which may explain why naturally occurring mutations are not as dominating as in the experiment. I wouldn't suggest that the authors make significant modification of the manuscript, which is already sufficiently comprehensive, but I think some elaboration in the discussion will be useful.

Finally, I wanted to make a note on the term "parasite" that describes Bartonella. In my knowledge, which might be outdated, parasites usually refer to eukaryotes but not prokaryotes. In the rodent-flea-Bartonella system, flea is usually considered as the (ecto)parasite, whereas Bartonella is usually considered as the pathogen / symbiont. If my question is valid, it will make sense if the authors justified the choice of terms (but does not need to change the term).

**Part II – Major Issues: Key Experiments Required for Acceptance**

Reviewer #1: 1. A major weakness of the study is the absence the biological relevance of TAA mutation has not been confirmed yet, although the authors found TAA inactivation showed higher bacterial loads in blood of Gerbillus addersoni only at day 5 post infection. Single time point difference cannot be defined as better fitness. Furthermore, the authors did not generate isogenic mutant strains (such as carrying a badE) to access a badE phenotype in rodent host.

2. Bacterial autotransporters are macromolecular proteins that play important roles in self aggregation, or adhesion with host cell matrix. Expression of such big proteins is very energy consuming. Therefore, it is not surprising to identify OFF mutations in these proteins when these proteins are not required under certain environment (as observed that BadA is inactivated when serially passaged on agar plates). In this manuscript, badE was inactivated during infection in Gerbillus andersoni while stayed invariable under infection in Gerbillus pyramidum, indicating that badE is required for establishment of infection in Gerbillus pyramidum. However, no difference was identified from the capacity of badE mutant to infect Gerbillus pyramidum against ancestor. This contradiction should be well addressed in the manuscript.

Reviewer #2: I have no major comments.

Reviewer #3: None.

**Part III – Minor Issues: Editorial and Data Presentation Modifications**

Reviewer #1: 1. Page9, line 182, “were presented”.

2. Page 21. Information of animals is too simple. Gerbillus is not a traditional lab animal. Are they adopted from wild environment or they are hand-fed? If they are captured from Negev desert, are they serum negative of Bartonella infection? It will be very important for the interpretation of the data.

Reviewer #2: Minor comments:

Pg 10 Fig 3 Would be nice to indicate males and females with different symbols.

Pg 17 Line 357-360. "Theoretical and empirical studies ....". Please had citations

Pg 27 Line 597 typo in “downsamled”

Reviewer #3: See above.

PLOS authors have the option to publish the peer review history of their article (what does this mean?). If published, this will include your full peer review and any attached files.

Reviewer #1: No

Reviewer #2: No

Reviewer #3: No

Figure Files:

Data Requirements:

Reproducibility:

References:

---

## [Decision Letter · Decision Letter 1]

13 Sep 2024

Dear Prof. Barrick,

We are pleased to inform you that your manuscript 'Pathogen contingency loci and the evolution of host specificity: Simple sequence repeats mediate *Bartonella* adaptation to a wild rodent host' has been provisionally accepted for publication in PLOS Pathogens.

Best regards,

Min Yue

Academic Editor

PLOS Pathogens

Francis Jiggins

Section Editor

PLOS Pathogens

Michael Malim

Editor-in-Chief

PLOS Pathogens

orcid.org/0000-0002-7699-2064

Reviewer Comments (if any, and for reference):

Reviewer's Responses to Questions

**Part I - Summary**

Reviewer #1: I have no more suggestions.

**Part II – Major Issues: Key Experiments Required for Acceptance**

Reviewer #1: (No Response)

**Part III – Minor Issues: Editorial and Data Presentation Modifications**

Reviewer #1: (No Response)

PLOS authors have the option to publish the peer review history of their article (what does this mean?). If published, this will include your full peer review and any attached files.

Reviewer #1: No

---

## [Editor Report · Acceptance letter]

20 Sep 2024

Dear Prof. Barrick,

We are delighted to inform you that your manuscript, "Pathogen contingency loci and the evolution of host specificity: Simple sequence repeats mediate *Bartonella* adaptation to a wild rodent host," has been formally accepted for publication in PLOS Pathogens.

Best regards,

Michael Malim

Editor-in-Chief

PLOS Pathogens

orcid.org/0000-0002-7699-2064